# *MaskTune*: Mitigating Spurious Correlations by Forcing to Explore

**Saeid Asgari Taghanaki\***
Autodesk AI Lab

**Aliasghar Khani\***
Autodesk AI Lab

**Fereshte Khani\***
Stanford University

**Ali Gholami\***
Autodesk AI Lab

**Linh Tran**
Autodesk AI Lab

**Ali Mahdavi-Amiri**
Simon Fraser University

**Ghassan Hamarneh**
Simon Fraser University

## Abstract

A fundamental challenge of over-parameterized deep learning models is learning meaningful data representations that yield good performance on a downstream task without over-fitting spurious input features. This work proposes *MaskTune*, a masking strategy that prevents over-reliance on spurious (or a limited number of) features. *MaskTune* forces the trained model to explore new features during a single epoch finetuning by masking previously discovered features. *MaskTune*, unlike earlier approaches for mitigating shortcut learning, does not require any supervision, such as annotating spurious features or labels for subgroup samples in a dataset. Our empirical results on biased MNIST, CelebA, Waterbirds, and ImagenNet-9L datasets show that *MaskTune* is effective on tasks that often suffer from the existence of spurious correlations. Finally, we show that *MaskTune* outperforms or achieves similar performance to the competing methods when applied to the selective classification (classification with rejection option) task. Code for *MaskTune* is available at https://github.com/aliasgharkhani/Masktune.

## 1 Introduction

Spurious correlations are coincidental feature associations formed between a subset of the input and target variables, which may be caused by factors such as data selection bias [Torralba and Efros, 2011, Jabri et al., 2016]. The presence of spurious correlations in training data can cause over-parameterized deep neural networks to fail, often drastically, when such correlations do not hold in test data [Sagawa et al., 2019] or when encountering domain shift [Arjovsky et al., 2019]. Consider the classification problem of cows and camels [Beery et al., 2018], where most of the images of cows vs. camels are captured on green fields vs. desert backgrounds due to selection bias (and perhaps the nature of the problem that camels are often in the desert). A model trained on such data may rely on the background as the key discriminative feature between cows and camels, thus failing on images of cows on non-green backgrounds or camels on non-desert backgrounds.

In over-parameterized regimes, there are often several solutions with almost identical loss values, and the optimizer (e.g., SGD) typically selects a low-capacity one [Wilson et al., 2017, Valle-Perez et al., 2018, Arpit et al., 2017, Kalimeris et al., 2019]. In the presence of spurious correlations, the optimizer might choose to leverage them as they generally demand less capacity than the expected semantic cues of interest, e.g., relying on the local color or texture of grass instead of the elaborate visual features that give a cow its appearance [Bruna and Mallat, 2013, Bruna et al., 2015, Brendel and Bethge, 2019, Khani and Liang, 2021].

---

*These authors contributed equally to this work.

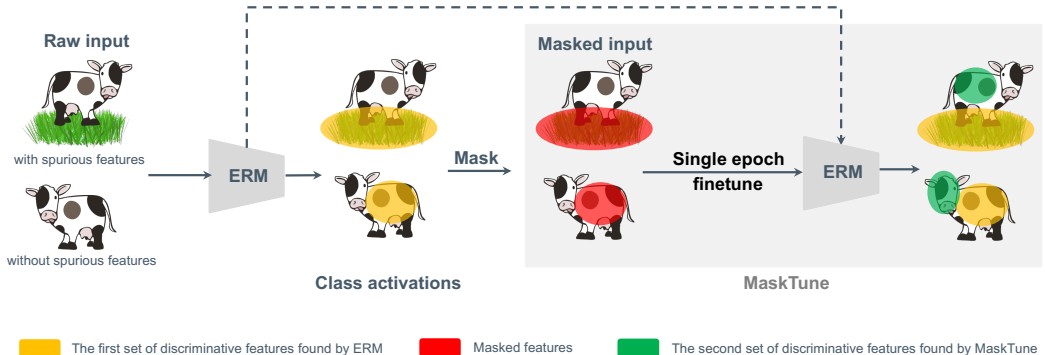

Figure 1: *MaskTune* generates a new set of masked samples by obstructing the features discovered by a model fully trained via empirical risk minimization (ERM). The ERM model is then fine-tuned for only one epoch using the masked version of the original training data to force new feature exploration. The features highlighted in yellow, red, and green correspond to features discovered by ERM, the masked features, and the newly discovered features by *MaskTune*, respectively.

In previous work, a supervised loss function has been employed to reduce the effect of spurious correlations [Sagawa et al., 2019]. However, identifying and annotating the spurious correlations in a large dataset as a training signal is impractical. Other works have attempted to force models to discard context and background (as spurious features) through input morphing using perceptual similarities [Taghanaki et al., 2021] or learning casual variables [Javed et al., 2020]. Discarding context and background, however, is incompatible with the human visual system that relies on contextual information when detecting and recognizing objects [Palmer, 1975, Biederman et al., 1982, Chun and Jiang, 1998, Henderson and Hollingworth, 1999, Torralba, 2003]. In addition, spurious features may appear on the object itself (e.g., facial attributes). Thus discarding the context and background may be a futile strategy in these cases.

Instead of requiring contextual and background information to be discarded or relying on a limited number of features, we propose a single-epoch finetuning technique called *MaskTune* that prevents a model from learning only the "first" simplest mapping (potentially spurious correlations) from the input to the corresponding target variable. *MaskTune* forces the model to explore other input variables by concealing (masking) the ones that have already been deemed discriminatory. As we finetune the model with a new set of masked samples, we force the training to escape its myopic and greedy feature-seeking approach and encourage exploring and leveraging more input variables. In other words, as the previous clues are hidden, the model is constrained to find alternative loss-minimizing input-target mappings. *MaskTune* conceals the first clues discovered by a fully trained model, whether they are spurious or not. This forces the model to investigate and leverage new complementary discriminatory input features. A model relying upon a broader array of complementary features (some may be spurious while others are not) is expected to be more robust to test data missing a subset of these features.

Figure 1 visualizes how *MaskTune* works via a schematic of the cow-on-grass scenario. Even in the absence of spurious correlations, models tend to focus on the shortcut (e.g., ears or skin texture of a cow), which can prevent models from generalizing to scenarios where those specific parts are missing. However, the object is still recognizable from the remaining parts. As an alternative, *MaskTune* generates a diverse set of partially masked training examples, forcing the model to investigate a wider area of the input features landscape e.g., new pixels.

A further disadvantage of relying on a limited number of features is the model's inability to know when it does not know. Let's go back to the cow-camel classification example; if cows only appear on grass in the training set then it is unclear which of the grass or the cow refers to the "cow" label. A model that only relies on the grass feature can confidently make a wrong prediction when some other object appear in the grass in the test time. We need the model to predict only if both cow and grass appear in the picture and *abstain* otherwise. One method used in the literature to address this issue is selective classification [Geifman and El-Yaniv, 2019, 2017, Khani et al., 2016], which allows a network to reject a sample if it is not confident in its prediction. Selective classification is essential in mission-critical applications such as autonomous driving, medical diagnostics, and

robotics as they need to defer the prediction to human if they are uncertain about the prediction. Learning different sets of discriminatory features, in addition to reducing the effect of spurious features, enables *MaskTune* to be applied to the problem of selective classification.

We apply *MaskTune* to two main tasks: a) robustness to spurious correlations, and b) selective classification. We cover four different datasets under (a) including MNIST with synthetic spurious features, CelebA and Waterbirds with spurious features in different subgroups [Sagawa et al., 2019], and the Background Challenge [Xiao et al., 2020] which is a dataset for measuring the reliance of methods on background information for prediction. Under (b) we test *MaskTune* on CIFAR-10 [Krizhevsky et al., 2009], SVHN [Netzer et al., 2011], and Cats vs. Dogs [Geifman and El-Yaniv, 2019] datasets. On both tasks, we outperform or perform similarly to the previous complex methods using our simple technique.

To the best of our knowledge, this is the first work to present a finetuning technique using masked data to overcome spurious correlations. Our contributions are summarized as follows:

1. We propose *MaskTune*, a new technique to reduce the effect of spurious correlations or over-reliance on a limited number of input features without any supervision such as object location or data subgroup labels.

2. We show that *MaskTune* leads to learning a model that does not rely solely on the initially discovered features.

3. We show how our method can be applied to selective classification tasks.

4. We empirically verify the robustness of the learned representations to spurious correlations on a variety of datasets.

## 2  Method

**Setup.** We consider the supervised learning setting with inputs $x \in \mathcal{X} \subset \mathbb{R}^d$ and corresponding labels $y \in \mathcal{Y} = \{1, \ldots, k\}$. We assume having access to samples $\mathcal{D}^0 = \{(x_i, y_i)\}_{i=1}^n$ drawn from an unknown underlying distribution $p_{data}(x, y)$.

Our goal is to learn the parameters $\theta \in \Theta$ of a prediction model $m_\theta : \mathcal{X} \to \mathcal{Y}$ that obtains low classification error w.r.t some loss function (e.g., cross entropy) $\ell : \Theta \times (\mathcal{X} \times \mathcal{Y}) \to \mathbb{R}$. Specifically, we minimize:

$$\mathcal{L}(\theta) = \mathbb{E}_{x,y \sim p_{\text{data}}(x,y)}[\ell(m_\theta(x), y)] \approx \frac{1}{n} \sum_{i=1}^n \ell(m_\theta(x_i), y_i) \tag{1}$$

where $n$ is the number of pairs in the training data.

Besides having good prediction accuracy, we aim to develop a model which does not solely rely on spurious or a limited number of input features. We propose to mask the input training data to create a new set. We then finetune (*for only one epoch*) a fully trained ERM model with the new masked data to reduce over-reliance on spurious or a limited number of features. The single epoch fine-tuning is done using a small learning rate e.g., the last decayed learning rate that the ERM has used in the first step. We found that large learning rates or more than one epoch fine-tuning leads to forgetting the discriminative features learned by the ERM.

**Input Masking.** A key ingredient of our approach is a masking function $\mathcal{G}$ that is applied offline (i.e., after full training). The goal here is to construct a new masked dataset by concealing the most discriminative features in the input discovered by a model after full training. This should encourage the model to investigate new features with the masked training set during finetuning. As for $\mathcal{G}$, we adopt the xGradCAM [Selvaraju et al., 2017], which was originally designed for a visual explanation of deep models by creating rough localization maps based on the gradient of the model loss w.r.t. the output of a desired model layer. Given an input image of size $H \times W \times C$, xGradCAM outputs a localization map $\mathcal{A}$ of size $H \times W \times 1$, which shows the contribution of each pixel of the input image in predicting the most probable class, i.e., it calculates the loss by choosing the class with highest logit value (not the true label) as the target class. After acquiring the localization map, for each sample $(x_i, y_i)$, where $x_i \in X$ and $y_i \in Y$, we mask the locations with the most contribution as:

$$\hat{x}_i = \mathcal{T}(\mathcal{A}_{x_i}; \tau) \odot x_i; \quad \mathcal{A}_{x_i} = \mathcal{G}(m_\theta(x_i), y_i) \tag{2}$$

where $\mathcal{T}$ refers to a thresholding function by the threshold factor $\tau$ (i.e., $\mathcal{T} = \mathbb{1}_{\mathcal{A}_{x_i} \leq \tau}$), and $\odot$ denotes element-wise multiplication. As the resolution of $\mathcal{A}$ is typically coarser than that of the input data, $\mathcal{T}(\mathcal{A}_{x_i})$ is up-sampled to the size of the input.

Procedurally, we first learn model $m_\theta^{\text{initial}}$ using original unmasked training data $\mathcal{D}^{\text{initial}}$. Then we use $m_\theta^{\text{initial}}$, $\mathcal{G}$ and $\mathcal{T}$ to create the masked set $\mathcal{D}^{\text{masked}}$. Finally, the fully trained predictor $m_\theta^{\text{initial}}$ is tuned using $\mathcal{D}^{\text{masked}}$ to obtain $m_\theta^{\text{final}}$.

As for the masking step, any explainability approach can be applied (note that some may have more computational complexity, such as ScoreCAM [Wang et al., 2020]). We use xGradCAM [Selvaraju et al., 2017] as it is fast and produces relatively denser heat-maps than other methods [Srinivas and Fleuret, 2019, Selvaraju et al., 2017, Wang et al., 2020].

## 2.1 *MaskTune* in Over-parameterized Regimes

Consider the overparametrized regime, in which the model family has sufficient complexity to fully fit the training data. It has been shown that deep neural nets are overparametrized and can fit completely random data [Zhang et al., 2021]. The generalization ability of deep neural nets is still not clear, but there are some speculation that connect the deep network generalization to their tendency of choosing simple functions that fit the training data [Valle-Perez et al., 2018, Arpit et al., 2017]. However, this simplicity bias can cause side effects such as their poor performance with respect to adversarial examples [Raghunathan et al., 2019] or to distribution shifts [Khani and Liang, 2021, Shah et al., 2020]. Here we study the effect of masking input features on complexity of a model in a situation where indeed the training procedure chooses the least complex model that fits training data. We show that in this case masking will result in learning a more complex model that discovers new features as the previous ones are blocked.

Formally, let $\mathcal{C}$ denote a function that measures model complexity and assume that the masking function $\mathcal{T}$ (as described in 2) only returns binary values, i.e., an indicator function that only keeps some features and zeros out the rest. We show that if training procedure returns the least complex model then masking results in a more complex model.

**Proposition 1.** *Consider an optimizing procedure that finds* $\min \mathcal{C}(m_\theta), s.t., \ell(m_\theta) = 0$ *as defined in 1. Let masking function* $\mathcal{T}$ *return binary values. If both models* $m_\theta$ *and* $m_\theta^{\text{initial}}$ *fit the training data (i.e., zero loss) then we have* $\mathcal{C}(m_\theta^{\text{final}}) \geq \mathcal{C}(m_\theta^{\text{initial}})$.

*Proof.* Note that both models belong to the model family ($m_\theta^{\text{initial}}, m_\theta^{\text{final}} \in \Theta$), and they both fit the training data. In the first step, training procedure chooses $m_\theta^{\text{initial}}$ over $m_\theta^{\text{final}}$; therefore according to our assumption $\mathcal{C}(m_\theta^{\text{final}}) \geq \mathcal{C}(m_\theta^{\text{initial}})$. $\square$

## 2.2 Adapting *MaskTune* for Selective Classification

Here we show how to use *MaskTune* for the selective classification problem. In order to make a more reliable prediction, we ensemble the original model ($m_\theta^{\text{initial}}$) and *MaskTune* ($m_\theta^{\text{final}}$) and only predict if both models agree. As a result, if there exist two sets of features that can predict the label, our method only predicts if both agree on the label (e.g., grass and cow in Figure 1).

To get an intuition on the performance of *MaskTune* for selective classification, similar to [Khani et al., 2016] we analyze the noiseless overparametrized linear regression. We show that *MaskTune* adaptation, explained above, abstains in the presence of covariate shift, thus leading to a more reliable prediction.

In particular, we show that *MaskTune* only predicts if the relationship between masked and unmasked features in training data holds in the test time. For example, if features describing "cow" are predictable of "grass" in training data (i.e., they always co-occur) then we only predict if they co-occur in the test data as well. Formally, let $s$ denote the masked feature after the first round and $z$ denote the rest of the $d - 1$ features. As we are in the overparametrized regime, $s$ can be predicted from $z$, let $\beta$ be the min-norm solution for predicting $s$ from $z$, i.e., $s = \beta^\top z$ for training data. We

now show that both models agree with each other for a new test set if $s = \beta^\top z$ in the test time as well. In other words, there is not covariate shift between $s$ and $z$.

**Proposition 2.** *Let $S \in \mathbb{R}^n$ be the concatenation, across all $n$ training samples, of a single masked scalar feature, out of $d$ possible features, and $Z \in \mathbb{R}^{n \times (d-1)}$ be the remaining $d - 1$ features. Let the model family $\Theta$ be the linear functions, and the optimization function chooses the min $L_2$-norm solution that fits the training data. The models trained with and without $S$ agree on the predicted output for a new test set $(z, s)$, iff $s = (Z^\top (ZZ^T)^{-1} S)^\top z$.*

The full proof is in the appendix. Note that $(Z^\top (ZZ^T)^{-1} S)$ is the min-norm solution for predicting $s$ from $z$ in the training data. This proposition states that the models agree only if the relation between the masked feature ($s$) and remaining features ($z$) in training holds in the test data as well.

In practice, we need to trade off between the coverage and precision. Therefore, instead of a hard threshold and predicting only if both models agree, we ensemble $m_\theta^{\text{initial}}$ and $m_\theta^{\text{final}}$ by multiplying their probabilities. In order to achieve the coverage goal, we find the desired threshold for abstaining in the validation set (see Section 4 for details).

## 3   Implementation Details

**Classification with Spurious Features.** For all of the datasets in Section 4.1, we use SGD with a momentum of 0.9 and weight decay of 1e-4. For biased MNIST, we used a simple convolutional neural network with four convolutional layers and a linear head. We trained this model with a batch size of 128 for 100 epochs with a learning rate of 0.01. We decreased the learning rate by a factor of 0.5 every 25 epochs. In the case of Background Challenge, we trained an ImageNet pre-trained ResNet50 with a batch size of 1,024 for 100 epochs decaying the learning rate by 0.1 after every 30 epochs. For the CelebA dataset, we trained an ImageNet pre-trained ResNet50 with a batch size of 512 for 20 epochs with a learning rate of 1e-4.

**Selective Classification.** For all the datasets in Section 4.2, we trained a ResNet32 from scratch. For CIFAR-10, we used SGD with a momentum of 0.9, weight decay of 1e-4, learning rate of 0.1, and batch size of 128. We trained it for 300 epochs and halved the learning rate every 25 epochs. For SVHN, we used the same hyperparameters as CIFAR-10 with only two differences: the weight decay and batch size were 5e-4 and 1,024, respectively. For Cats vs. Dogs dataset, we used Adam optimizer with a weight decay of 1e-4 and a learning rate of 0.001. We trained the model for 200 epochs with a batch size of 128 and dropped the learning rate by a factor of 0.1 on epoch 50.

**Masking Threshold.** Our goal is to mask the most important features, i.e., the core of the heat maps generated by the explainability methods. Masking a few input variables has almost no effect on the model's behavior, whereas large masks may destroy useful signals in the input, resulting in very low training accuracy. To reduce the search space of the masking threshold $\tau$ across all tasks, we experimented with $\tau = \{\mu_i, \mu_i + \sigma_i, \mu_i + 2\sigma_i, \mu_i + 3\sigma_i\}$ where $\mu_i$ and $\sigma_i$ represent mean and standard deviation over the heatmap values for training sample $x_i$. We also experimented with mean value over *all* training samples, using soft masks (i.e., no threshold), and sorting and masking the K-top activated variables. We found $\mu_i + 2\sigma_i$ works better in general as $\mu_i$ and $\mu_i + \sigma_i$ remove a large portion of the input while $\mu_i + 3\sigma_i$ removes very few variables.

## 4   Experimental Results

We evaluate our *MaskTune* method on two main applications: a) classification with spurious correlations—we expect *MaskTune* to prevent such correlations by identifying and masking them and b) selective classification where the ability to abstain is critical—we expect *MaskTune* to improve reliability by forcing a model to investigate additional variables in the input and learn more complex relationships. In each experiment, we compare our method to relevant baselines and competing methods.

### 4.1   Classification with Spurious Features

**MNIST with Colored Squares on the Corners:** As a warm-up, we test the ability of our method to distinguish between two MNIST digit groups (0-4 and 5-9) in the presence of spurious features.

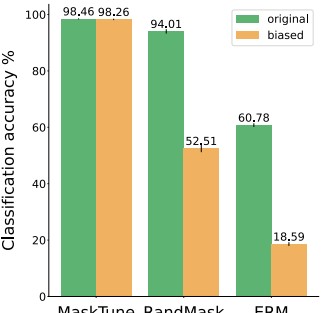
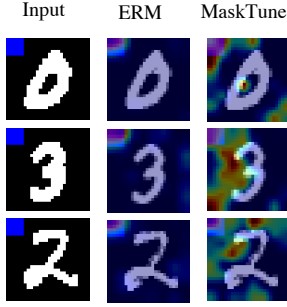

Figure 2: Left: Test accuracy on the original and biased MNIST datasets. *MaskTune* outperforms other methods on both the original and biased test sets. Right: xGradCAM visualizations before (ERM) and after applying *MaskTune*. *MaskTune* enforces exploring more input features, which leads to more robust predictions. Here the spurious feature is the blue square on the top left corner.

We construct a dataset such that a classifier would achieve poor accuracy by relying on spurious input features. To this end, we group digits labeled 0 to 4 into class 0 and those labeled 5 to 9 into class 1. Next, in the training set, we place 99% and 1% of the new class 0 and new class 1 data on a background with a small blue square on the top left corner, respectively, and keep the remaining data intact. We use two test sets during testing: the original raw MNIST test set and a biased test set. To create the biased test set we place all of the digits from the group "5-9" on a background with a small blue square (representing spurious features) on the top left corner and keep all of the digits from the group "0-4" unchanged (i.e., we don't add any squares to their background and use the original images). Figure 2 demonstrates the performance of the ERM, RandMask, and our *MaskTune* on both original and biased test sets. The RandMask method is similar to *MaskTune*, but masks a *randomly* chosen area of the input image. As shown in Figure 2, *MaskTune* outperforms other methods by a large margin on both test sets. As demonstrated in Figure 2 (right), *MaskTune* forces the model to explore more features, as opposed to the ERM, which only looks into the spurious feature (the blue square). We also ran an experiment with multiple spurious features (Appendix **??**) and reported results for iterative version of *MaskTune*.

**Classification with Spurious Features in Subgroups.** In this experiment, we leverage the CelebA [Liu et al., 2015] and the Waterbirds [Sagawa et al., 2019] datasets. In the CelebA dataset, there is a high correlation between features gender={male, female} and hair_color={blond, dark}, meaning that the feature gender might be used as a proxy to predict the hair_color. In other words, an ERM would assign the label dark to male images since the majority of male images have dark hair, and there are only 1,387 (0.85%) blond males in the training set of size 162,770. To balance the accuracy of predictions on different subgroups of data, the existing methods [Sagawa et al., 2019] use subgroup information, i.e., wrong predictions of an ERM on the worst group (blond males) can be penalized during training. Some other approaches [Levy et al., 2020, Nam et al., 2020, Pezeshki et al., 2021, Taghanaki et al., 2021, Liu et al., 2021] use subgroup information during model selection (labeled validation set). However, it is impractical to recognize "all" subgroups in a dataset and label them. In Table 1, we show that *MaskTune* achieves comparable performance to both these groups of methods *without* using group information in training or in model selection. We also show *MaskTune* significantly improves worst-group accuracy (78% vs. 55% classification accuracy) in comparison to methods that do not use subgroup information during training or model selection. In Figure 3 (left), we highlight the important input features for predicting hair color on the CelebA dataste. As shown, the ERM model leverages gender features while *MaskTune* forces to investigate other features as well.

The Waterbirds dataset [Sagawa et al., 2019] was proposed to assess the degree to which models pick up spurious correlations in the training set. We discovered and fixed two *issues* with the Waterbirds dataset: a) because the background images in the Places dataset [Zhou et al., 2017] may already contain bird images, multiple birds may appear in an image after overlaying the segmented bird images from the Caltech-UCSD Birds-200-2011 (CUB) dataset [Wah et al., 2011]. For example, the label of an image may be "landbird", but the image contains both land and water birds. We manually removed such images from the dataset. b) Because the names of the species are similar, some land

Table 1: Results from the CelebA dataset using ResNet-50. Our method outperforms all fully unsupervised methods.

| Method | Group labels in training set | Group labels in validation set | Worst-group accuracy | Avgerage accuracy |
|---|---|---|---|---|
| Group DRO Sagawa et al. [2019] | Yes | Yes | **88.3** | 91.8 |
| CVaR DRO Levy et al. [2020] | No | Yes | 64.4 | 82.5 |
| LfF Nam et al. [2020] | No | Yes | 77.2 | 85.1 |
| SD Pezeshki et al. [2021] | No | Yes | **83.2±2.0** | 91.6±0.6 |
| JTT Liu et al. [2021] | No | Yes | 81.1 | 88.0 |
| CIM Taghanaki et al. [2021] | No | Yes | 81.3 | 89.2 |
| ERM Sagawa et al. [2019] | No | No | 47.2 | 95.6 |
| CVaR DRO Levy et al. [2020] | No | No | 36.1 | 82.5 |
| LfF Nam et al. [2020] | No | No | 24.4 | 85.1 |
| JTT Liu et al. [2021] | No | No | 40.6 | 88.0 |
| DivDis Lee et al. [2022] | No | No | 55.0 | 90.8 |
| *MaskTune* (ours) | No | No | **78.0±1.2** | 91.3±0.1 |

Figure 3: Activation visualizations of ERM and *MaskTune* for CelebA (left) and Waterbirds (right) samples. *MaskTune* enforces exploring new features. As demonstrated, after applying *MaskTune*, the task-relevant input signals (hair colour and bird features) are emphasised.

birds have been mislabeled as waterbirds which we corrected. The *corrected* Waterbirds dataset can be found on *MaskTune*'s GitHub page. After addressing the two issues, the ERM model's worst-group accuracy increased from 60% which is reported in [Sagawa et al., 2019] to 80.8±1.3%. We repeated the group-DRO method and ERM experiments on the corrected waterbirds dataset and reported the results in Table 2. As demonstrated, our method (without any group supervision) achieves similar accuracy to the group-DRO that benefits from full supervision. In Figure 3 (right), we visualized feature importance before and after applying our *MaskTune* on the modified Waterbirds dataset.

**The Background Challenge.** As a further step, we evaluated *MaskTune* on the Background Challenge data [Xiao et al., 2020] to see if the positive observations from the MNIST experiment apply to a

Table 2: Results from the Waterbirds dataset using ResNet-50. Our method significantly improves ERM's worst-group accuracy without supervision.

| Method | Group labels in training set | Group labels in validation set | Worst-group accuracy | Avgerage accuracy |
|---|---|---|---|---|
| GroupDRO [Sagawa et al., 2019] | Yes | Yes | **89.3±3.1** | 94.4±0.7 |
| ERM | No | No | 80.8±1.3 | 94.0±0.2 |
| *MaskTune* | No | No | **86.4±1.9** | 93.0±0.7 |

more realistic scenario. The Background Challenge is a publicly available dataset that consists of ImageNet-9 [Deng et al., 2009] test sets with various levels of foreground and background signals. It is intended to assess how much deep classifiers rely on spurious features for image classification. We used two configurations to compare *MaskTune*'s performance: Only FG, in which the background is completely removed, Mixed-same, in which the foreground is placed on a different background from the same class, and Mixed-rand, where the foreground is overlaid onto a random background.

As shown in Table 3, *MaskTune* outperforms the baseline ResNet-50's performance by 2.5% on the Only-FG test set and 1.2% on Mixed-same, showing that *MaskTune* does not rely much on the background and uses both background and foreground for prediction. On the Mixed-same and Only-FG test sets, *MaskTune* outperforms other techniques because mixing/removing the background texture/info confuses other methods. These results show that our technique helps to learn task-relevant features without depending on nuisance signal sources.

Table 3: Results from the Background Challenge on ImageNet-9 using ResNet-50. Our method outperforms the baselines on both Mixed-same and Only FG test sets.

| Method | Original | Mixed-same | Mixed-rand | Only-FG |
|---|---|---|---|---|
| Baseline [Xiao et al., 2020] | 96.3 | 89.8 | 75.6 | 85.6 |
| CIM [Taghanaki et al., 2021] | **97.7** | 89.8 | **81.1** | - |
| SIN [Sauer and Geiger, 2021] | 89.2 | 73.1 | 63.7 | - |
| INSIN [Sauer and Geiger, 2021] | 94.7 | 85.9 | 78.5 | - |
| INCGN [Sauer and Geiger, 2021] | 94.2 | 83.4 | 80.1 | - |
| *MaskTune* (Ours) | 95.6 | **91.1** | 78.6 | **88.1** |

## 4.2 Selective Classification

For selective classification task, we evaluated *MaskTune* on three datasets: CIFAR-10 [Krizhevsky et al., 2009], SVHN [Netzer et al., 2011], and Cats vs. Dogs [Geifman and El-Yaniv, 2019], with coverage values of {90%, 95%, 100%}. For example 90% coverage means abstaining 10% of the samples.

Given input image $j$ and the original ($m_\theta^{\text{initial}}$) and finetuned ($m_\theta^{\text{final}}$) models, let their respective inference-time prediction probabilities for class $i$ be $P_{ij}^{\text{init}}$ and $P_{ij}^{\text{final}}$. Then *MaskTune* does not abstain and declares the class $i$ as the predicted class, iff $P_{ij}^{\text{init}} \cdot P_{ij}^{\text{final}} > \gamma$, where $\gamma$ is a threshold that allows for controlling the target coverage. To find the proper $\gamma$ for achieving the targeted coverage, we iterate over the validation set and find a threshold that, if applied to the validation set, would give the desired coverage (i.e., if our target coverage is 90%, we seek for a threshold which if we apply for abstention on the validation set, we will abstain predicting 10% of validation data). Note that the probabilities of the two models are multiplied to ensure that as the value of either probability decreases, the possibility of abstention increases.

We compare our method to the other selective classification approaches such as Softmax Response (SR) [Geifman and El-Yaniv, 2019], SelectiveNet (SN) [Geifman and El-Yaniv, 2019], Deep Gamblers (DG) [Liu et al., 2019], and One-sided Prediction (OSP) [Gangrade et al., 2021]. Like the OSP, we first split the training data into train and validation sets. After training, we search for the abstention threshold on the validation set, then use it at the test time. As seen in Table 4, *MaskTune* outperforms all prior techniques on all three datasets on 95% and 100% coverage rates. For results on more coverage rates refer to Appendix **??**.

Table 4: Selective classification results on CIFAR-10, SVHN, and Cats vs. Dogs datasets for different coverage values.

| Dataset | Target Coverage | SR Cov. | SR Err. | SN Cov. | SN Err. | DG Cov. | DG Err. | OSP Cov. | OSP Err. | MaskTune Cov. | MaskTune Err. |
|---|---|---|---|---|---|---|---|---|---|---|---|
| Cifar-10 | 100% | 99.99 | 9.58 | 100 | 11.07 | 100 | 10.81 | 100 | 9.74 | 99.99±0.02 | **8.96±0.48** |
| | 95% | 95.2 | 8.74 | 94.7 | 8.34 | 95.1 | 8.21 | 95.1 | 6.98 | 94.86±0.18 | **6.54±0.39** |
| | 90% | 90.5 | 6.52 | 89.6 | 6.45 | 90.1 | 6.14 | 90.0 | **4.67** | 89.73±0.22 | 4.74±0.31 |
| SVHN | 100% | 99.97 | 3.86 | 100 | 4.27 | 100 | 4.03 | 100 | 4.27 | 100.0±0.00 | **3.68±0.16** |
| | 95% | 95.1 | 1.86 | 95.1 | 2.53 | 95.0 | 2.05 | 95.1 | **1.83** | 95.19±0.09 | 1.84±0.23 |
| | 90% | 90.0 | 1.04 | 90.1 | 1.31 | 90.0 | 1.06 | 90.1 | 1.01 | 89.55±0.26 | **0.96±0.11** |
| Cats vs. Dogs | 100% | 100 | 5.72 | 100 | 7.36 | 100 | 6.16 | 100 | 5.93 | 99.98±0.00 | **4.83±0.17** |
| | 95% | 95.0 | 3.46 | 95.2 | 5.1 | 95.1 | 4.28 | 95.1 | 2.97 | 95.01±0.14 | **2.96±0.15** |
| | 90% | 90.0 | 2.28 | 90.2 | 3.3 | 90.0 | 2.5 | 90.0 | **1.74** | 90.78±0.16 | 1.94±0.18 |

## 5 Related Work

In the following, we review the related works which have attempted to mitigate the effect of spurious correlation in training deep models. We also discuss previous methods that use attention-based online masking and how our method differs.

**Robustness to Spurious Correlations.** Distributionally robust optimization (DRO) [Ben-Tal et al., 2013, Gao et al., 2017, Duchi et al., 2021] has been proposed to improve generalization to worst cases (minority distributions) in a dataset. However, the DRO objective leads to disproportionate attention to the worst cases, even if they are implausible. To address this issue, Sagawa et al. [2019] proposed to leverage subgroup information during optimization. Although this method reduces the likelihood of the worst-case failure, it is based on prior solid information (i.e., subgroup labels), which is not always available. Several efforts have been made to reduce the subgroup-level supervision. Sohoni et al. [2020] proposed a clustering-based method for obtaining sub-group information to be used in DRO setup. However, determining the number of clusters (sub-groups) in a dataset is not trivial since a small number may still dismiss minor subgroups while a large number may lower overall accuracy. Yaghoobzadeh et al. [2019] proposed measuring sample accuracy throughout training to discover forgettable instances, then fine-tuning models using such samples to increase model resilience against spurious correlations. Chen et al. [2020] demonstrated that self-training can also help decrease the effect of spurious features, but only if the source classifier is very accurate and there are not too many isolated sub-groups in the data. Liu et al. [2021] discovered that training a model twice helps it become resistant to spurious correlation. However, if a model reaches high classification accuracy in the first run, this technique is no longer useful since there is not enough misclassified data to retrain the model. *MaskTune*, on the other hand, employs an orthogonal approach to learn robust representations through gradient-based masking and fine-tuning and does not require subgroup-level labels during training or model selection.

**Online Attention-based Masking.** A large number of studies suggest using attention-based (soft) masks to eliminate irrelevant information from input data during training [Sharma et al., 2015, Wang et al., 2017, Xu et al., 2015, Zheng et al., 2017]. Although these methods increase overall classification accuracy, they are incapable of disregarding spurious correlations since attention may be on an area of the input where the spurious correlations are dominant during training. For concentrating attention only on the foreground, Li et al. [2018] developed a guided attention model. Their technique, however, needs additional annotations of object locations/masks. Nonetheless, none of these methods aimed to reduce the effects of spurious correlations. Even with complete supervision (i.e., masking the whole background), spurious correlations might still occur in the foreground. *MaskTune*, on the other hand, produces masks from a fully trained model and uses them for a single epoch fine-tuning. *MaskTune* does not require any additional supervision such as object location annotations.

# 6  Conclusion

In this work, we considered the problem of preventing models from learning spurious correlations. We introduced a new fine-tuning technique that is designed explicitly for spurious correlations. It enforces a model to explore more variables in the input and map them to the same target. Through experiments and theoretical analysis on classification with nuisance background information which typically suffers from the presence of spurious correlations in the data, we showed that models trained with *MaskTune* outperform previous relevant methods. We also showed that *MaskTune* helps to improve the accuracy significantly in selective classification tasks. Adapting *MaskTune* for non-image data, such as sentiment analysis, can be an intriguing future work. Another area for future work is *how* best to identify and use the most effective masked samples (rather than all) when fine-tuning the final model, perhaps using uncertainty information or active learning. Although *MaskTune* can help reduce the effect of many types of spurious correlation, such as texture, color, and localized nuisance features e.g., artifacts added to x-ray images by medical imaging devices, there are some cases where *MaskTune* may not be effective, such as a small transformation in all pixel values for some of the images in a dataset. This occurs in medical devices or cameras that add almost imperceptible color (values) to captured images.

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
