# OpenReview forum: "MaskTune: Mitigating Spurious Correlations by Forcing to Explore"
_NeurIPS.cc/2022/Conference — NeurIPS 2022 Accept_

### Official Review · Reviewer_eFV8 · 2022-06-14

**Rating:** 6
**Confidence:** 3
**Soundness:** 3 good
**Presentation:** 3 good
**Contribution:** 3 good

**Summary:**

This paper proposes a simple technique to avoid reliance on "spurious correlations": 1) train a model, 2) fine-tune for an additional epoch, but, mask the part of the data that is deemed important for making the predictions. This forces the classifier to "explore" and learn something beyond the initial "spurious correlation". Results on e.g., Waterbirds and CelebA, the method outperform other methods which don't use group information.

**Questions:**

I do not expect that all of these questions are answered.
- What happens in the synthetic MNIST experiment when there are two spurious correlations. E.g., two squares instead of one?
- Why only fine-tune for one epoch? What happens when you do more?
- Does MaskTune help on natural distribution shifts such as ImageNet -> ImageNetV2.
- What happens when there is no spurious correlations, does MaskTune reduce accuracy? What happens if you run MaskTune on the benchmark ImageNet dataset.

**Limitations:**

This is marked as Yes in the checklist but a limitations section could be appreciated in the revision.

**Strengths And Weaknesses:**

Strengths:
The method is straightforward and, for the most part, very intuitive. Moreover, the performance is validated empirically on multiple datasets, where MaskTune (the method) outperforms other methods which do not have access to group information and approaches the performance of methods which do.

Weaknesses:
The main concern is that this method is only tested on datasets where spurious correlations exist. E.g., I am not surprised the MNIST case works. My concern is that, if I don't know apriori that spurious correlations exist, should I apply MaskTune or not? Are spurious correlations really a problem on datasets like ImageNet? Currently it is unclear if I should apply this method when training on ImageNet. It is even possible that MaskTune could reduce accuracy when training on datasets without spurious correlations, as the important features could be masked out.

---

> ### Author Response · Authors · 2022-08-02
> **More clarifications**
>
> - *What happens in the synthetic MNIST experiment when there are two spurious correlations. E.g., two squares instead of one?*
>
> Please see the general response section
>
> - *Why only fine-tune for one epoch? What happens when you do more?*
>
> By finetuning for more epochs, the model starts forgetting the previously learned features which might lead to overall performance drop.
>
> - *Does MaskTune help on natural distribution shifts such as ImageNet -> ImageNetV2.*
>
> Thank you for the suggestion. We will test our method on ImageNetV2. An interesting future development will be to investigate how MaskTune can be modified to take into account the natural distribution shifts.
>
> - *What happens when there is no spurious correlations, does MaskTune reduce accuracy? What happens if you run MaskTune on the benchmark ImageNet dataset.*
>
> We ran MaskTune on ImageNet. The overall test accuracy did not decrease significantly i.e., from 78.862 (ERM) to 78.213 (MaskTune). After applying MaskTune classification accuracy on certain classes improved significantly e.g., by over 50%. For example for class “projectile, missile” the accuracy was improved from 25% to 80%, for class “crane” from 44% to  82%, for class “skunk, polecat, wood pussy” from 49% to 69%, for class “coffeepot” from 48% to 64%, etc.

---

> > ### Comment · Reviewer_eFV8 · 2022-08-04
> > **Thank you for the response.**
> >
> > Thanks for the response, I continue to recommend acceptance.

---

### Official Review · Reviewer_5uk4 · 2022-07-09

**Rating:** 6
**Confidence:** 4
**Soundness:** 3 good
**Presentation:** 3 good
**Contribution:** 3 good

**Summary:**

The authors propose a two-stage pipeline ('MaskTune') for mitigating spurious correlations where the the first stage simply entails training a convolutional model via ERM. GradCAM is is then applied to the resulting model to generate salient maps; these are converted into masks by thresholding and used to mask out the inputs during the second stage wherein the model is fine-tuned for a single epoch, the goal being to force the model to  explore alternative views of the data.  Experiments are preformed on a range of image dataset for both regular classification and selective classification, in which the option to abstain from predicting is added.  The authors propose to ensemble the models obtained from the first and second stages of training to obtain the probabilities used for the latter task, optimising the threshold to meet the desired coverage using a validation set. The proposed method performs competitively with other recent domain generalisation methods without needing the groups/environments/domains to be annotated, as is the case for popular methods such as GDRO.


**Questions:**

- How robust is the chosen thresholding heuristic? Do you expect it to hold in general and can you suggest a more rigorous approach to setting it for if not (one which is based on something more concrete than avoiding masking too many/too few variables)?
- Is a two-stage process, whereby spurious features are identified based on only a single round of training, sufficient in general to avoid the learning of spurious correlations? It seems that a dataset consisting of multiple sets of features that spuriously correlate with the target (for instance, background colour and background texture) would pose problems for MaskTune.

**Limitations:**

I am satisfied with the extent to which the societal impacts of the method are addressed -- the interpretability of the work could be highlighted to further bolster the paper in this respect. The authors address a couple of limitations of the method -- it's unimodality and sample inefficiency -- at the end of the conclusion, however I would like to see the the discussion on this topic expanded, both with regard to those limitations already noted and in the regard to addressing some of those concerns such as those raised in "Strengths are Weaknesses"/"Questions" (or to see relevant ablations provided if in fact those concerns are in fact unjustified)

**Strengths And Weaknesses:**

# Strengths
- Despite the similarity in spirit to pre-existing method (which are appropriately cited) the method nonetheless appears to be novel and is appealing for  its relative simplicity compared with other domain generalisation methods.
That said, I worry that it might be too brittle to be practically useful, for several reasons:
1) From what I gather, the efficacy of the method is highly dependent on the choice of masking threshold, $\lambda$ -- choosing a value that is too low would allow for leakage of the spurious features, while, as discussed in the paper, choosing a value that is too low leads to portions of the samples being masked out, including those salient features which we hope to learn during the fine-tuning phase. The authors propose to mask using a heuristic based on (standard) deviations from the mean, however the justification for it feels a bit hand-wavy. Moreover, the method relies heavily on the efficacy of the mask-generation method and it seems artefacts induced by upscaling (which may be significant depending on the choice of architecture and the input resolution) could prove detrimental.
2) It seems likely that in many cases several a single iteration of the method would be insufficient, as -- to continue with the motivating example adduced in the paper -- one can imagine only a portion of the background features being needed to reliably distinguish between cows and camels in the training set, resulting in other background features remaining unmasked during fine-tuning. One could repeat the procedure until some stopping criterion is reached but it's not obvious whether problems such as catastrophic forgetting begin to factor into the optimisation process, hindering the efficacy of MaskTune.
3) If there are no spurious correlations in the training set and the set of unmasked features contain no information predictive of the class label there may be risk of an over-parameterised model resorting to memorisation -- that we only fine-tune for a single epoch may mean that this typically isn't a problem, but there nonetheless seems like such a risk is there with the current incarnation of the algorithm.
4) Dovetailing from the previous point, a single epoch of fine-tuning may not be a sufficient duration for learning more complex sets of features/unlearning the bias learned during the first stage of training, yet training training for longer may also carry its share of problems (such as memorisation/catastrophic forgetting).
The advantage of the method, however, is that the above problems can, in theory, be readily identified -- this demands an auditing process however, one that is likely to be time-consuming, especially if multiple iterations are required

 - Figure 1 provides a straightforward illustration of the process, however it would be useful to have a legend
 denoting the meaning of different mask colourings.

- Comparison with other works is sufficiently thorough -- highlighting the distinction between the proposed method
and Just train Twice (JtT), which adopts a similar two-stage approach (and how the latter can fail) being particularly pertinent.

- Good range of datasets and a good number of relevant baselines which cover both standard classification and selective classification,
- While currently listed as an avenue of future work, I think how one MaskTune might be concretely adapted to other
tasks (e.g, segmentation) and modalities warrants some discussion.

- Indications  table indicating what sources of supervision are required for the methods run on the Waterbirds dataset is nice to see
 but exactly what is meant by model-selection supervision isn't explained ('Train Supervision' should also perhaps be indicated to be w.r.t. the group labels) -- it's meaning can be inferred but I don't think it's self-evident to forgo explanation (even if only at the caption level_.

- The paper is for the most part easily understandable and soundly structured. However there are some areas of the text which could be clearer; the area which perhaps stood out the most in this respect was that relating to the biased MNIST setup, with the sentence "We place 99% and 1% of all digits labelled "0-4" and "5-9" in the training  set on a background with a small blue square on the top left corner, respectively, and keep the remaining data intact" being quite difficult parse. On the topic of this section, it would also be useful to have Figure 2 include a visual description of the two setups considered.

- Error intervals (whether these correspond to standard error or deviation is not stated) are given
for only a subset of the methods and datasets (none of the results for selective classification
have them, for instance).

- I found it odd that in explaining the ensembling and threshold-selection procedure for selective classification
(paragraph beginning line 241) the authors choose to introduce new notation for the initial and final models when
notations different from that established in Section 2 of the paper.

---

> ### Author Response · Authors · 2022-08-02
> **More clarifications**
>
> Thank you for the suggestions on improving Fig 1 and writing. We will fix these in the final version.
>
> - *Error intervals (whether these correspond to standard error or deviation is not stated) are given for only a subset of the methods and datasets (none of the results for selective classification have them, for instance).*
>
> The paper that we used to report other methods' accuracy, only reports the accuracy (not the error intervals). But we report the error intervals for ours in table 5 in appendix. We will move Table 5 to the paper from the appendix.
>
> - *I found it odd that in explaining the ensembling and threshold-selection procedure for selective classification (paragraph beginning line 241) the authors choose to introduce new notation for the initial and final models when notations different from that established in Section 2 of the paper.*
>
> We will fix this in the final version
>
>
> - *How robust is the chosen thresholding heuristic? Do you expect it to hold in general and can you suggest a more rigorous approach to setting it for if not (one which is based on something more concrete than avoiding masking too many/too few variables)?*
>
> We previously tried several different thresholding strategies:
>
> 1- mean over all training samples (this is too strong)
>
> 2- chose top K number of highly activated pixels (here K needs to be tuned)
>
> 3- soft masks (i.e., just normalize the heatmaps between 0 and 1 and do not threshold)
>
> 4- mean over each image
>
> 5- mean + 2std over each sample
>
> 6- mean + 3std over each sample
>
> We found #5 works in general better across all datasets.
>
> - *Is a two-stage process, whereby spurious features are identified based on only a single round of training, sufficient in general to avoid the learning of spurious correlations? It seems that a dataset consisting of multiple sets of features that spuriously correlate with the target (for instance, background colour and background texture) would pose problems for MaskTune.*
>
> Please see the general response section.
>
> We will expand the conclusion section by elaborating more on the limitations of our work such as what kinds of spurious features might be missed by our method, running times, etc. and future research opportunities.

---

### Official Review · Reviewer_dGKz · 2022-07-11

**Rating:** 7
**Confidence:** 4
**Soundness:** 3 good
**Presentation:** 3 good
**Contribution:** 3 good

**Summary:**

Spurious learning or shortcut learning problem has received increasing attention from the community recently. This work suggests the masking approach MaskTune, in order to avoid relying too heavily on spurious (or a small number of) features. More specifically, MaskTune forces the trained model to explore new features during a single epoch finetuning by masking previously discovered features. Experimental results show that the proposed method could improve the worst-group accuracy for spurious learning tasks. In addition, it also achieves decent performance for the selection classification task.


**Questions:**

Please refer to the Strengths And Weaknesses part

**Limitations:**

Yes

**Strengths And Weaknesses:**

Advantages:
1. Spurious correlation is a problem worth studying. The idea of encouraging the model to explore new features is sound.
2. The paper is overall well written.
3. Experimental results validate that the proposed method can be used to alleviate the reliance on spurious correlations and be applied in the selection classification task.


Some minor concerns:
1. The positions of sections 3 and 4 should be switched. Implementations should be introduced before experimental results.
2. Experimental evaluation is only performed using toy datasets, such as MNIST and CelebA. It is unclear whether the proposed method could be applied in more challenging real-world datasets such as those from the WILDS benchmark [1].
3. The writing can be improved. In the abstract and the introduction section, the authors mentioned the selection classification task and did not give too much introduction to this task. Only after reading Section 2.2, the readers can understand what is the selection classification task. This could be unfriendly for readers not familiar with this task.
4. On line 107, the authors mention that one major benefit of using xGradCAM is that it could produce dense heatmaps. What would happen if sparse heatmaps are generated?

[1] Koh, Pang Wei, Shiori Sagawa, Henrik Marklund, Sang Michael Xie, Marvin Zhang, Akshay Balsubramani, Weihua Hu et al. "Wilds: A benchmark of in-the-wild distribution shifts." In International Conference on Machine Learning, pp. 5637-5664. PMLR, 2021.

---

> ### Author Response · Authors · 2022-08-02
> **More clarification**
>
> - *The positions of sections 3 and 4 should be switched. Implementations should be introduced before experimental results.*
>
> Thank you for pointing this out. We will make this change in the final version.
>
> - *Experimental evaluation is only performed using toy datasets, such as MNIST and CelebA. It is unclear whether the proposed method could be applied in more challenging real-world datasets such as those from the WILDS benchmark [1].*
>
> We have tested on the WaterBirds dataset (Table 2) and the Background challenge (Table 3) as well. Our method has significantly improved the baselines on both the datasets.  We ran our method on ImageNet. The overall test accuracy did not decrease significantly i.e., from 78.862 (ERM) to 78.213 (MaskTune), but after applying MaskTune classification accuracy on certain classes improved significantly e.g., by over 50%. For example for class “projectile, missile” the accuracy was improved from 25% to 80%, for class “crane” from 44% to 82%, for class “skunk, polecat, wood pussy” from 49% to 69%, for class “coffeepot” from 48% to 64%, etc.
>
> - *The writing can be improved. In the abstract and the introduction section, the authors mentioned the selection classification task and did not give too much introduction to this task. Only after reading Section 2.2, the readers can understand what is the selection classification task. This could be unfriendly for readers not familiar with this task.*
>
> We agree. We will fix this by adding more background info in the introduction on selective classification.
>
> - *On line 107, the authors mention that one major benefit of using xGradCAM is that it could produce dense heatmaps. What would happen if sparse heatmaps are generated?*
>
> This is an important point. In comparison to masking only the core of a dense map, sparse heatmaps often mask larger areas of the input. Because there isn't much information left in the input after masking a lot of pixels, learning the second best discriminative features is impossible.

---

### Official Review · Reviewer_tJb1 · 2022-07-11

**Rating:** 7
**Confidence:** 4
**Soundness:** 3 good
**Presentation:** 3 good
**Contribution:** 3 good

**Summary:**

This paper focuses on the important problem of mitigating reliance on spurious correlations. In particular, the paper proposes MaskTune, a masking strategy to prevent over-reliance on spurious correlations. The masking strategy forces the model to reduce relaince on spurious correlations and learn new features via the masked examples. The empirical results show that MaskTune improve model performance on multiple datasets and has applications to the task of selective classification.

**Questions:**

Please see strengths and weaknesses

**Limitations:**

Please see strengths and weaknesses

**Strengths And Weaknesses:**

**Strengths**

- Well-written: The problem is well-motivated, figure 1 does a good job at describing the masking approach, the experiment setup, datasets, and results are easy to follow.
- The proposed masking approach to spread reliance across multiple features is conceptually simple + effective and does not require knowledge of contextual / background information.
- The empirical results on mitigating spurious correlations are significantly better than existing unsupervised methods when you do not have access to supervision at model selection time (a more realistic setting than what has been considered previously)
- The connection between spurious correlations (narrow subset of features) and selective classification is quite interesting; the paper presents initial theoretical results in the overparameterized linear regression setup and the ensemble idea is empirically better than three existing methods on three datasets.

**Weaknesses**

- Baselines: the randmask baseline is interesting but quite underutilized in the experiments on real datasets. Also, the paper does not provide any details about this method (e.g., how aggressive is the random cropping? how many copies of the same image with different random masks are used?).  The gap between randmask and masktune seems a bit too significant even for the toy MNIST dataset and I am wondering if a stronger version of randmask can be used in these experiments.
- Saliency maps: The masktune approach heavily relies on explanation / saliency maps working “as expected”. However, feature attribution methods in general are hard to evaluate and work on evaluation of interpretability methods have shown that these methods have major failure modes.  There is work [R1, R2, R3] that indicate that saliency maps are ineffective at detecting (and so mitigating) spurious correlations. A discussion of this issue and about the usage of a specific saliency map method (xGradCAM) is missing. In particular, how robust are these results to a different choice of explaination method (on realistic datasets such celebA and waterbirds)?
- Spurious correlations and locality: it seems like this method would only work in cases where the spurious correlation can be localized in input space. That is, the spurious correlation “region” in the image is different from the “region” of the desirable / core feature. For example, in the MNIST experiment, if instead of adding a patch, I add a red vs. blue tint to the entire image, the model will just learn the color feature. Will masktune mitigate shortcut learning in this case? It would be great if the authors can shed light on this.
- Are the number in table 3 and 4 averaged over multiple runs or is it best-of-K? are the differences / improvements statistically significant?

[R1] Post hoc Explanations may be Ineffective for Detecting Unknown Spurious Correlation

[R2] Do Input Gradients Highlight Discriminative Features?

[R3] Sanity Simulations for Saliency Methods

---

> ### Author Response · Authors · 2022-08-02
> **More clarifications**
>
> - *Baselines: the randmask baseline is interesting but quite underutilized in the experiments on real datasets. Also, the paper does not provide any details about this method (e.g., how aggressive is the random cropping? how many copies of the same image with different random masks are used?). The gap between randmask and masktune seems a bit too significant even for the toy MNIST dataset and I am wondering if a stronger version of randmask can be used in these experiments.*
>
> We ran another version of a random masking method with different window sizes: On MNIST data, for each image, we randomly selected a  window of {2x2, 3x3, 4x4, .... nxn} pixels where n is the image size divided by 2. As you can see [here](https://imgur.com/Jp2rEB7), MaskTune still outperforms the random masking approach.
>
>
> - *Saliency maps: The masktune approach heavily relies on explanation / saliency maps working “as expected”. However, feature attribution methods in general are hard to evaluate and work on evaluation of interpretability methods have shown that these methods have major failure modes. There is work [R1, R2, R3] that indicate that saliency maps are ineffective at detecting (and so mitigating) spurious correlations. A discussion of this issue and about the usage of a specific saliency map method (xGradCAM) is missing. In particular, how robust are these results to a different choice of explaination method (on realistic datasets such celebA and waterbirds)?*
>
> Thank you for the question.  According to our experiments, several explainability (saliency) methods would work well if masking thresholds were properly set, i.e., not all masking methods work with the same threshold. ScoreCAM, for example, would result in similar performance to xGradCAM, but we found ScoreCAM to be very slow because it needs to run on all the 2D feature maps at the end of the network, i.e., the network should run 2048 times per sample in the ResNet50 case. We agree that explainability methods may be ineffective for detecting certain types of spurious correlations (R1, R2, R3). However, by masking initially discovered discriminative features, our MaskTune leads to the exploration of more features, even if they are not spurious.
>
> - *Spurious correlations and locality: it seems like this method would only work in cases where the spurious correlation can be localized in input space. That is, the spurious correlation “region” in the image is different from the “region” of the desirable / core feature. For example, in the MNIST experiment, if instead of adding a patch, I add a red vs. blue tint to the entire image, the model will just learn the color feature. Will masktune mitigate shortcut learning in this case? It would be great if the authors can shed light on this.*
>
> Please see the general response section
>
>  - *Are the number in table 3 and 4 averaged over multiple runs or is it best-of-K? are the differences / improvements statistically significant?*
>
> To be consistent with other works, in Table 4 we reported the accuracy, but in the appendix (Table 5), we have the results with standard deviation over 3 runs. The standard deviation over 3 runs is small. We will replace Table 5 in appendix with Table 4 in the paper and report standard deviation for Table 3 as well.

---

### Author Response · Authors · 2022-08-02
**General Response**

We value the reviewers' incisive observations and compliments on the novelty of our work and clarity of writing. We begin with three general comments and then address the specific comments made by each reviewer. We will incorporate all of the reviewers' recommendations for minor writing edits, improving/adding figures, and other changes into the final version.

**1- Is it harmful to use MaskTune when we are unsure whether there is a spurious correlation in our data?**

Even if there are no spurious features in the input or detecting the spurious feature is difficult, MaskTune masks the initially found most discriminative features, resulting in the discovery of the second discriminative set of features. For example, if an ERM model initially finds the cow's head to be the most discriminative feature, masking it (which is not spurious) forces the model to find the second best feature as well, which could be the texture of the cow's skin. The final model would learn both the cow's head and skin texture as discriminative features, rather than just the head.


**2- Does MaskTune handle all types of spurious correlations?**

Although Masktune can help reduce the effect of many types of spurious correlation, such as texture, color, and localized nuisance features e.g., artifacts added to x-ray images by medical imaging devices, there are some cases where MaskTune may not be effective, such as when a small transformation is applied to all pixel values of some of the images in a dataset. This occurs in medical devices or cameras that add almost imperceptible color (values) to captured images.

**3- What happens if we have more than one spurious feature?**

Running MaskTune for only one iteration on a dataset with more than one spurious feature still performs better than ERM. We tested on MNIST. Please see the results [here](https://imgur.com/pm13NYc) . If we use MaskTune iteratively, the performance improves even more. We ran two iterative versions of MaskTune. We added two coloured patches to MNIST digits as two distinct spurious features. We ran 1, 2, and 3 iterations of masking. The method works well when we do iterative accumulative masking (i.e., add new masks to the previously masked samples), but when we apply masks from each iteration to the raw input(non-accumulative), the results are not as good as the accumulative version. To reduce the running time, we only used one iteration of masking. One way to stop the accumulative masking is to monitor the training accuracy. If the model is not able to fit the data after N masking iterations because there are no useful features left, we can stop.

---

### Meta-Review · Area_Chair_GEeC · 2022-08-31

**Recommendation:** Accept
**Confidence:** Certain

**Metareview:**

The paper considers the problem of reliance of NN models to spurious correlations and proposes MaskTune a method to alleviate this. MaskTune is a masking strategy to prevent over-reliance on spurious correlations. The masking strategy forces the model to reduce reliance on spurious correlations and learn new features via the masked examples. The empirical results show that MaskTune improve model performance on multiple datasets and has applications to the task of selective classification.

The paper focuses on a very important current problem with DNN models, and the proposed idea works on a number of benchmarks. The paper is well-written too. There were a number of questions raised by the reviewers and additional experiments requested that were addressed by the authors during the rebuttal period. Therefore I vote for acceptance and I ask the authors to update the paper accordingly for the camera ready version.

**Award:**

No

---

### Decision · Program_Chairs · 2022-09-14

Accept